# Mito-TEMPO Improves the Meiosis Resumption and Mitochondrial Function of Vitrified Sheep Oocytes via the Recovery of Respiratory Chain Activity

**DOI:** 10.3390/ani14010152

**Published:** 2024-01-02

**Authors:** Xi Zhao, Airixiati Dilixiati, Luyao Zhang, Aikebaier Aihemaiti, Yukun Song, Guodong Zhao, Xiangwei Fu, Xuguang Wang, Abulizi Wusiman

**Affiliations:** 1Department of Animal Science, College of Animal Science, Xinjiang Agricultural University, Urumqi 830052, China; 2State Key Laboratory of Animal Biotech Breeding, China Agricultural University, Beijing 100193, China

**Keywords:** sheep, oocytes, Mito-TEMPO, mitochondria, vitrification

## Abstract

**Simple Summary:**

Vitrification causes oxidative stress and organelle damage in oocytes. The results of this study reveal that Mito-TEMPO can efficaciously improve the oxidative stress injury of vitrified oocytes by recovering mitochondrial function via the mitochondrial respiratory chain. In this experiment, we supplemented GV-stage oocyte vitrified thawing and in vitro maturation with Mito-TEMPO to observe its effects on sheep oocyte vitrification. It was shown that Mito-TEMPO could not only rescue the apoptosis and mitochondrial dysfunction of vitrified oocytes but also regulate their intracellular Ca^2+^ homeostasis and ATP content, as well as provide strong antioxidant properties. Our analysis of single-cell transcriptomes revealed that the positive impact of Mito-TEMPO on vitrified oocytes is facilitated by the respiratory chain, which indicates that Mito-TEMPO may be a promising mitochondria-targeted antioxidant to enhance vitrified oocyte quality.

**Abstract:**

Vitrification is a crucial method for preserving animal germ cells. Considering the increased oxidative stress and organelle damage incurred, it is still necessary to make the process more efficient for oocytes. As the energy source of oocytes, mitochondria are the most abundant organelle in oocytes and play a crucial role in their maturation. Here, we found that Mito-TEMPO, a mitochondria-targeted antioxidant, could efficaciously improve the oxidative stress injury of vitrified oocytes by recovering mitochondrial function via the mitochondrial respiratory chain. It was observed that Mito-TEMPO not only improves oocyte viability and meiosis but also maintains spindle structure. A subsequent study indicated that Mito-TEMPO effectively rescued mitochondrial dysfunction and attenuated vitrification-induced oxidative stress. Further investigation revealed that Mito-TEMPO regulates vitrified oocytes’ intracellular Ca^2+^ homeostasis and ATP content and provides strong antioxidant properties. Additionally, an analysis of the transcriptome at the single-cell level revealed that the respiratory chain mediates the beneficial effect of Mito-TEMPO on vitrified oocytes. Overall, our findings indicate that supplementing oocytes with Mito-TEMPO is an effective method to shield them from the damage caused by vitrification. In addition, the beneficial effects of Mito-TEMPO on vitrified sheep oocytes could inspire further investigations of the principles underlying oocyte cryobiology in other animals.

## 1. Introduction

In recent years, vitrification has replaced slow freezing as the most efficient technique for preserving female fertility. The technology has broad applications, including assisted reproductive technology (ART), the preparation of genetic banks, and the preservation of endangered species. Vitrification is distinguished from slow freezing by its ability to prevent the formation of intracellular and extracellular ice [1]. However, the standardization and improvement of oocyte quality is required, as it is affected by impairment caused by vitrification or slow freezing. The decline in oocyte quality following vitrification can be attributed to the excessive production of reactive oxygen species (ROS) [2]. This phenomenon impacts essential cell functions and leads to apoptosis, DNA damage, and mitochondrial dysfunction [3]. Conversely, exposure to high concentrations of cryoprotectants, such as dimethyl sulfoxide (DMSO) and ethylene glycol (EG), can result in cytotoxicity. A previous study provided evidence of the cytotoxic effects of cryoprotective additives through the use of Raman microspectroscopy. The study demonstrated that exposure to DMSO and EG led to a changed protein distribution in the cortex and a notable disruption to the cortical F-actin network in ovine oocytes [4]. Recently, approaches to mitigate cytotoxicity have involved elevating the viscosity of vitrification media by incorporating macromolecular supplements [5,6], using microdevices to minimize ice nucleation and crystal growth [7], as well as introducing antifreeze proteins [8].

Mitochondria, as the primary energy-producing organelles in the oocyte cytoplasm, are essential for oocyte maturation, fertilization, and subsequent embryonic development [9]. Mitochondria play a crucial role in supplying adenosine triphosphate (ATP) to facilitate various cellular and molecular processes [10]. Mitochondrial dysfunction has been identified as a significant cause of ROS [11]. Furthermore, this phenomenon has also been observed in aged animals and individuals with metabolic disorders following mitochondrial replacement therapy [12], antioxidant treatment, or treatment targeting mitochondria [13]. As a result, targeting and reducing mitochondrial oxidative damage is an effective treatment strategy for preserving the viability of cumulus-oocyte complexes (COCs).

Triphenyl phosphonium chloride (Mito-TEMPO, MT) is a mitochondria-targeted antioxidant, acting as a superoxide dismutase mimetic and antioxidant within the mitochondria [14]. In previous studies, Mito-TEMPO has been used to treat diseases caused by mitochondrial dysfunction [15]. Moreover, Mito-TEMPO has also been reported in in vitro embryo culture [16,17] and in investigations on semen cryopreservation [18,19,20,21]. It is reported that supplementation with 0.1 μM Mito-TEMPO decreased oxidative stress and damage and restored the meiotic maturation of COCs and expansion of CCs that were affected by pre-treatment with bisphenol A [17]. Mito-TEMPO also enhanced the maturation potential of bovine COCs [16]. In cryopreservation studies, the use of 50 μM Mito-TEMPO enhanced post-thaw sperm quality in rams [18], humans [19], roosters [20], and sheep [21] by reducing ROS production and enhancing antioxidant capacity. There is a shortage of recent research on the use of Mito-TEMPO as mitochondria-targeted antioxidants in oocyte cryopreservation.

The developmental rates in sheep oocytes vitrification are still poor in both GV and MII-stage oocytes. At present, the primary aim is to improve the efficiency of sheep oocytes vitrification and the quality of oocytes after vitrification. Here, we discovered that supplementing vitrified GV-stage sheep COCs with Mito-TEMPO can restore mitochondrial function and meiosis defects to enhance oocyte maturation captivity. Furthermore, Mito-TEMPO avoided the formation of excessive ROS and maintained calcium homeostasis. Through analyzing the single-cell transcriptome of oocytes, it was further confirmed that the addition of Mito-TEMPO enhanced the quality of vitrified oocytes via ATP transportation and the mitochondrial respiratory chain.

## 2. Materials and Methods

### 2.1. Experimental Design

Experiment 1: The effect of Mito-TEMPO on the in vitro maturation and mitochondrial function of vitrified sheep COCs. In this experiment, the collected COCs were randomized into a fresh group, a vitrification group (vit group), and a supplement Mito-TEMPO vitrification group (vit + MT group). In the vit + MT group, different concentrations of Mito-TEMPO (0, 0.5, 1, and 5 μM) were added to vitrification, thawing, and in vitro maturation media. The optimal concentration of Mito-TEMPO was determined by measuring cell survival and the rate of PBE. After determining the optimal concentration of Mito-TEMPO, the study further investigated spindle structure, redox status, mitochondrial function, and Ca^2+^ levels. A total number of 1500 oocytes in at least three replications were included in this experiment.

Experiment 2: Transcriptomic analysis of the fresh, vit, and vit + MT groups. After in vitro maturation, 30 oocytes were picked up from each group. The differentially expressed genes (DEGs), Gene Ontology (GO), and Kyoto Encyclopedia of Genes and Genomes (KEGG) pathway enrichment were analyzed. Additionally, eight randomly selected DEGs were quantified using RT-qPCR to verify them. A total number of 90 oocytes in three replications were included in this experiment.

Experiment 3: The potential effect of Mito-TEMPO on sheep oocyte mitochondrial function was studied. In this experiment, the optimal concentration of Mito-TEMPO obtained in Experiment 1 was added to the maturation medium of unvitrified COCs, which was named the MT group. After IVM, PBE rate and mitochondrial function (including ∆Ψm, mitochondria distribution, and ATP content) were measured from fresh and MT groups, respectively. A total number of 322 oocytes in at least three replications were included in this experiment.

### 2.2. Collection of COCs

Ovine ovaries were purchased from a local slaughterhouse (Hetian, China) and stored in a thermos flask containing a preheated saline solution at 30 °C while being transported to the lab within 2 h. The COCs were aspirated using a 10 mL syringe from 3 to 8 mm follicles. The follicular fluid containing the COCs was poured into a 90 mm dish containing oocyte washing medium consisting of HEPES-buffered TCM 199 (TCM199-Hepes, Gibco, Waltham, MA, USA) supplemented with 2% *v*/*v* fetal bovine serum (FBS, Sigma, Saint Louis, MO, USA) and 0.1% heparin sodium (Sigma, USA). Under a light microscope (Olympus, Tokyo, Japan), those COCs exhibiting a minimum of two to three compact layers of CCs and a homogeneous cytoplasm were chosen.

### 2.3. Vitrification and COC Warming

The COCs were vitrified using the open pulled-straws method. Briefly, the COCs underwent a triple rinse in base medium (BM; TCM199-Hepes supplemented with 20% FBS) and were then transferred into equilibration solution (10% *v*/*v* EG (Sigma, USA) and 10% *v*/*v* DMSO (Sigma, USA)) for 5 min. Five COCs were transferred into the vitrification solution (20% *v*/*v* EG and 20% *v*/*v* DMSO in 0.5 mol/L sucrose) for 30 s. The vitrified COCs were stored in liquid nitrogen (LN_2_) for at least 1 week. For thawing, the COCs were rinsed in 0.5, 0.25, and 0.15 mol/L sucrose for 1 min, 5 min, and 5 min, respectively, and then rinsed three times in BM. All the solutions were kept at 38.5 °C.

### 2.4. In Vitro Maturation

The COCs were washed three times in maturation medium (TCM199 with Earle salts; Gibco, USA) supplemented with 10% FBS, 2 IU/mL FSH (Solarbio, Beijing, China), 0.5 IU/mL LH (Solarbio, China), 1 μg/mL 17β-estradiol (Sigma, USA), 0.3 mM sodium pyruvate, 100 µM cysteamine, and 50 μg/mL gentamicin. A group of 40–50 COCs were then transferred into 4-well dishes (Thermo, Waltham, MA, USA) containing 600 μL maturation medium covered with mineral oil (Sigma, USA). All dishes were prewarmed for at least 2 h in an incubator at 38.5 °C under 5% CO_2_ in air before use. Then, the COCs were cultured in the medium for 24 h under the same conditions.

### 2.5. Oocyte Survival and the First Polar Body Extrusion (PBE)

The survival rate of oocytes was referenced by Succu et al. [22]. Briefly, after in vitro maturation, the oocytes were morphologically evaluated immediately under a light microscope (Olympus, Japan). Oocytes with faint cytoplasm or a damaged membrane and/or zona pellucida were considered not viable. Then, the extrusion of the first polar body of the surviving oocyte was observed under a light microscope (Olympus, Japan).

### 2.6. Immunofluorescence (IF) Staining

Denuded matured oocytes underwent fixation using 4% (*w*/*v*) paraformaldehyde (PFA) for a duration of 45 min at room temperature (RT). Subsequently, the oocytes were made permeable using 0.5% Triton X-100 (Solarbio, China) at RT for an hour. After being blocked in 3% BSA solution for 1 h at RT, the oocytes were reacted with anti-α-tubulin-FITC antibody, (Sigma, USA, 1:2000) overnight at 4 °C. Then, they were washed with 0.1% Tween 20 three times. Finally, the oocytes were dyed with 4′,6-diamidino-2-phenylindole (DAPI, Vector Laboratories, Burlingame, CA, USA) for 6 min, and the fluorescent images were taken with laser scanning confocal microscopy (Leica, Wetzlar, Germany).

### 2.7. Intracellular ROS and GSH Level Assay

According to a manuscript in a previous article [23], the denuded matured oocytes were placed into 1 mmol/L 2′,7′-dichlorofluorescein diacetates (2′,7′-DCFHDA, Invitrogen, Carlsbad, CA, USA) to assess ROS or into 10 μmol/L Cell Tracker Blue (Invitrogen, USA) to detect GSH at 38.5 °C in 5% CO_2_ for 30 min. After that, the oocytes were thoroughly washed with PBS at least three times. By utilizing the same exposure time, the intensity of fluorescence was measured using a confocal laser scanning microscope with a FITC filter for ROS and a DAPI filter for GSH excitation, respectively. The mean fluorescence intensity per oocyte was analyzed by ImageJ software (Version 1.48; National Institutes of Health, Bethesda, MD, USA).

### 2.8. Quantification of Mitochondrial Function Staining

For active mitochondrion staining and mitochondrion ROS, the oocytes were incubated in 5 nM cell-permeant MitoTracker GreenFM (Invitrogen, USA) or MitoSOX Red (Invitrogen, USA) mitochondrial superoxide indicator for 20 min at 38.5 °C and kept in a dark place with 5% CO_2_ in the air. After washing three times with PBS, the oocytes were mounted on nonfluorescent glass slides and observed under the laser scanning confocal microscope. We referred to the method of Stojkovic et al. to judge the mitochondrial normal and abnormal distribution [24]. For MitoSOX, the mean fluorescence intensity per oocyte was quantified by using the same confocal microscope parameters. The intensity of fluorescence was assessed using ImageJ software.

### 2.9. Quantification of Mitochondrial Membrane Potentials (∆Ψm)

To measure the ∆Ψm, the oocytes were assessed using a JC-1 assay kit (Solarbo, Beijing, China). First, mature denuded oocytes were placed in a 10 mmol/L JC-1 at 38.5 °C with 5% CO_2_ for 25 min. Subsequently, the oocytes underwent a triple rinse with PBS and were observed using the same confocal microscope parameters via a laser scanning confocal microscope. Mean fluorescence intensity of each oocyte was quantified using ImageJ software. The calculation of ∆Ψm was used to determine the ratio of red, indicative of activated mitochondria (J-aggregates), to green fluorescence, which represents less active mitochondria (J-monomers).

### 2.10. ATP Content Determination

The ATP content in oocytes was quantified utilizing the Enhanced ATP Assay Kit (Beyotime, Shanghai, China). According to previous research [25], the ATP standards were diluted in a series ranging from 0 to 40 pmol. Matured oocytes from each group were denuded and collected in 20 μL of lysis buffer and then vortexed until the cells were fully lysed. Next, the well was incrementally infused with standard solutions, ATP assay buffer, and ATP detection diluent buffer. The EnSpire microplate reader (PerkinElmer, Waltham, MA, USA) was used to measure luminescence activity promptly. ATP content was calculated using a standard curve.

### 2.11. Quantification of Cytoplasmic Ca^2+^ ([Ca^2+^]_c_), Mitochondrial Ca^2+^ ([Ca^2+^]_m_), and Endoplasmic Reticulum Ca^2+^ ([Ca^2+^]_ER_)

Subcellular Ca^2+^ levels were assessed using Rhod 2-AM (Invitrogen, USA), Fluo 4-AM (Invitrogen, USA), and Fluo 3-AM (Invitrogen, USA) to measure the [Ca^2+^]_m_, [Ca^2+^]_ER_, and [Ca^2+^]_c_, respectively. All of these dyes are living cell stains. The oocytes were incubated in a prewarmed dye for 30 min at 38.5 °C with 5% CO_2_. Then, the oocytes were triple rinsed with PBS and observed using the same confocal microscope parameters, utilizing a laser scanning confocal microscope. The mean fluorescence intensity per oocyte was assessed using Image J software.

### 2.12. Building a Single-Cell RNA Library and Sequencing the Transcriptome

The transcriptomic analysis of oocytes was carried out using a single-cell RNA-Seq procedure. Concisely, within the lysis buffer, each group had three specimens gathered (10 oocytes in each sample). The sequencing of the developed library was conducted using the Illumina platform. The initial sequencing data (Raw Reads) obtained from Hiseq sequencing were processed to remove low-quality sequences and adapter contamination. This resulted in the generation of high-quality sequences (Clean reads) used for all subsequent analyses. RNA integrity was evaluated using the RNA Nano 6000 Assay Kit on the Bioanalyzer 2100 system (Agilent Technologies, Santa Clara, CA, USA). mRNA was purified from total RNA using poly-T oligo-attached magnetic beads. Fragmentation was performed using divalent cations at elevated temperatures in the First Strand Synthesis Reaction Buffer (5×). At last, the PCR products were purified (AMPure XP system), and the quality of the library was evaluated using the Agilent Bioanalyzer 2100 system. The index-coded samples were clustered using a cBot Cluster Generation System with a Tru-Seq PE Cluster Kit v3-cBot-HS (Illumina, San Jose, CA, USA), following the manufacturer’s instructions. After cluster generation, library preparations were sequenced on an Illumina Nova-seq platform, and 150 bp paired-end reads were generated.

### 2.13. RNA Isolation and Real-Time Quantitative PCR (RT-qPCR)

Total RNAs were extracted using Transzol reagent (Transgen, Beijing, China) and reverse transcribed using TransScript One-Step gDNA Removal and cDNA Synthesis SuperMix (Transgen, China). Table 1 enumerates the documented primers utilized for the reference RNA sequences in the real-time quantitative polymerase chain reaction studies. Efficiency tests were conducted on the primers to verify their specificity. The RT-qPCR process involved mixing 1 μL of cDNA with PerfectStrat Green qPCR superMix (Transgen, China), adding both forward and reverse primers (10 μmol/L), and RNase-free water to reach a total volume of 20 μL. A ViiA7 Real-Time PCR System (Applied Biosystems, NY, USA) was used to quantify the relative abundance of specific transcripts. The cycling conditions were 95 °C for 2 min, followed by 40 cycles of 94 °C for 15 s, 50 or 55 °C for 30 s, and 72 °C for 30 s. The relative mRNA quantities of the target genes were determined using the 2^−ΔΔCt^ method, with GADPH representing the reference genes.

### 2.14. Statistical Analysis

All percentage or values derived from a minimum of three iterations of the experiments, presented as a mean ± SEM, are denoted in parentheses as (n). The data were analyzed using a paired samples *t*-test with GraphPad Prism 6 statistical software. *p* < 0.05 was used as the criterion for significant differences.

## 3. Results

### 3.1. Mito-TEMPO Improved the Oocyte Viability and Meiotic Maturation after Vitrification

The COCs were randomly divided into fresh, vit, and vit + MT groups (Figure 1a). We first focused on assessing the survival rates of the vitrified-thawed oocytes with Mito-TEMPO after maturation. Different concentrations of Mito-TEMPO (0, 0.5, 1, and 5 μM) were added to vitrification, thawing, and in vitro maturation media. Figure 1c illustrates that 1 μM Mito-TEMPO enhanced the survival rate of the vitrified-thawed oocytes (75.01 ± 3.33% vs. 62.64 ± 2.52%, *p* < 0.05). The analysis of the PBE rate showed that the meiotic resumption in the presence of 1 μM Mito-TEMPO was superior to that of the vitrified group (Figure 1d). Then, we selected a concentration of 1 μM Mito-TEMPO to conclude our research. Considering that the primary cause of meiotic arrest was a flawed spindle structure, this was further investigated across all groups. Immunostaining revealed a significant increase in abnormal spindle formation in the vitrified oocytes relative to the controls, yet this was mitigated using Mito-TEMPO (Figure 1e,f). These results suggest that Mito-TEMPO can maintain the standard spindle structure to improve the maturation ability of vitrified oocytes.

### 3.2. Mito-TEMPO Inhibits Oxidative Stress after Vitrification

Excess levels of reactive oxygen species may lead to decreased oocyte quality and lower recovery rates after vitrification. Consequently, the levels of ROS and GSH in the oocytes were subsequently assessed. As shown in Figure 2, Mito-TEMPO can reduce the ROS level (29.21 ± 0.57 vs. 31.50 ± 0.54, *p* < 0.01) and increase the GSH level (111.37 ± 3.56 vs. 56.52 ± 4.25, *p* < 0.0001) to alleviate the oxidative stress in oocytes induced by vitrification.

### 3.3. Mito-TEMPO Restores Mitochondrial Function in Vitrified Oocytes

In order to confirm the enhancement of mitochondrial performance in vitrified oocytes by Mito-TEMPO, we first studied the mitochondria distribution pattern using Mito-Tracker. Our findings revealed a clustering of numerous mitochondria within the cytoplasm of vitrified oocytes (Figure 3a). Quantitatively, more than 65% of vitrified oocytes showed mislocalized mitochondria, and Mito-TEMPO decreased this proportion to 40% (Figure 3b). We also used MitoSOX probes to quantify mitochondrial ROS levels. As shown in Figure 3a,c, Mito-TEMPO reduced the vitrification-induced overproduction of mitochondrial ROS (26.17 ± 1.70 vs. 44.60 ± 2.97, *p* < 0.0001).

Our observations of irregular mitochondrial distribution and elevated ROS levels suggest the potential impairment in their functionality due to vitrification; we then assessed ∆Ψm using JC-1 (Figure 3d). Compared with the fresh group, the vitrified oocytes showed a reduced red-to-green ratio (1.23 ± 0.98 vs. 1.44 ± 0.37, *p* < 0.05) but an increase in Mito-TEMPO vitrified oocytes (1.74 ± 0.06 vs. 1.23 ± 0.98, *p* < 0.0001, Figure 3e).

Furthermore, mitochondria are the prominent organelle of ATP production. So, we measured the ATP content of oocytes in each group. We observed a significant reduction in ATP content in the vitrified oocytes (4.88 ± 0.23 vs. 9.43 ± 0.46, *p* < 0.01) compared to fresh oocytes, and this reduction was reversed by Mito-TEMPO (8.93 ± 0.23 vs. 4.88 ± 0.23, *p* < 0.01, Figure 3f).

Altogether, these results suggest that Mito-TEMPO can ameliorate vitrified-induced mitochondrial dysfunction in oocytes.

### 3.4. Mito-TEMPO Restores Mitochondrial and Cytoplasmic Ca^2+^ Levels in Vitrified Oocytes

Mitochondria are the energy powerhouse of the oocyte and play a significant role in cellular Ca^2+^ signaling, which is crucial for oocyte viability and is involved in the dynamic regulation of cellular Ca^2+^ [26]. Therefore, we investigated the calcium levels in oocytes. A previous study showed that Rhod 2-AM and Fluo 4-AM could accurately track mitochondria Ca^2+^ ([Ca^2+^]_m_) and endoplasmic reticulum Ca^2+^ ([Ca^2+^]_ER_) in oocytes, respectively [27]. We also utilized Fluo 3-AM to analyze the cytoplasmic calcium ([Ca^2+^]_c_) concentration. As depicted in Figure 4, the [Ca^2+^]_m_ (18.01 ± 0.49 vs. 15.38 ± 0.74, *p* < 0.01) showed a significant increase, while the [Ca^2+^]_c_ (51.27 ± 2.77 vs. 62.02 ± 2.44, *p* < 0.01) exhibited a significant decrease after vitrification. Nonetheless, the concentration of [Ca^2+^]_ER_ remained unchanged after vitrification. Predictably, Mito-TEMPO successfully reinstated the [Ca^2+^]_m_ (15.59 ± 0.69 vs. 18.01 ± 0.49, *p* < 0.05) and [Ca^2+^]_c_ levels (59.23 ± 2.04 vs. 51.27 ± 2.77, *p* < 0.05) in the vitrificated oocytes.

### 3.5. Investigating the Target Effectors of Mito-TEMPO in Vitrified Oocytes Using Single-Cell Transcriptome Analysis

In order to gain a better understanding of the impact of Mito-TEMPO on vitrified oocyte quality, we conducted a single-cell transcriptome analysis on the IVM oocytes from the fresh, vit, and vit + MT groups. The heatmap and volcano plot indicated a variance in the transcriptome profile between the vit and fresh groups (Figure 5a–c). In the vitrified oocytes, 354 DEGs were downregulated, while 700 DEGs were upregulated (Figure 5b). In addition, the vit + MT group showed 88 downregulated DEGs and 173 upregulated DEGs compared to vitrified oocytes (Figure 5c). Then, we analyzed the DEGs of three groups and identified 41 genes in common (Figure 5d). Quantitative real-time PCR confirmed the co-expression of various randomly selected DEGs (Figure 5e).

The GO analysis of the DEGs showed that genes enriched in the BMP signaling pathway, ATP synthesis-coupled electron transport, translation, and ATP synthesis-coupled proton transport were abnormally expressed in the vitrified oocytes when compared with the fresh oocytes, but this was recovered in the vit + MT group (Figure 6). In addition, cellular component analysis showed genes associated with mitochondrion and respiratory chain complex IV. KEGG revealed the presence of genes enriched in oxidative phosphorylation, thermogenesis, metabolic pathways, and fat acid metabolism in the vitrified oocytes but this was restored by Mito-TEMPO (Figure 7). Collectively, these various biological pathways and processes show a strong correlation with mitochondrial function.

### 3.6. Mito-TEMPO at a Concentration of 1 μM Had No Potential Toxic Effect on Oocyte Mitochondrial Function

In order to examine the potential toxic impact of 1 μM Mito-TEMPO on the mitochondrial function of sheep oocytes, we added 1 μM Mito-TEMPO to fresh COCs and assessed their mitochondrial function. The findings show that the PBE rate in 1 μM Mito-TEMPO was significantly higher than in the nonsupplemented group (Figure 8a). Meanwhile, ∆Ψm (Figure 8b,c), the percentage of normal mitochondria distribution (Figure 8d), and the ATP content (Figure 8e) in the oocytes showed similar trends. These results suggest that Mito-TEMPO at a concentration of 1 μM has a beneficial effect on oocyte mitochondrial function.

## 4. Discussion

In this study, we investigated the impact of Mito-TEMPO on mitochondrial function and the mechanism by which Mito-TEMPO improved the viability and meiotic maturation of vitrified sheep GV-stage COCs. As we know, this study represents the first attempt to apply mitochondrial-targeted antioxidants in an oocyte cryopreservation model and elucidate the protective effects of Mito-TEMPO over mitochondrion during vitrification.

The most immediate effect of vitrification on GV-stage oocytes is a decrease in germinal vesicle breakdown and polar body extrusion [28,29]. Furthermore, GV-stage sheep oocytes are more susceptible to damage during vitrification compared to other developmental stages [30]. Sheep oocytes have a high amount of intracellular lipids, second only to porcine oocytes; this phenotype results in the increased production of H_2_O_2_ and oxidative stress [31]. Our findings show that oocyte survival and maturation were significantly impacted by vitrification. Mito-TEMPO increased PB extrusion in vitrified oocytes and had a positive impact on spindle formation. Similarly, previous research has demonstrated that therapies targeting mitochondria can reverse the spindle defects caused by oxidative stress in oocytes from mice and humans [32].

The endogenous antioxidant systems are altered in vitrified oocytes, resulting in an increase in the level of ROS [33]. Thus, various antioxidants were used to enhance the antioxidant capacity of the culture environment and decrease the accumulation of ROS in vitrification cells [27,34,35]. Our findings confirmed that vitrification damaged the mitochondria of GV-stage oocytes and increased the production of ROS, which is consistent with previous studies. After treatment with 1 μM Mito-TEMPO, the abnormal distribution of mitochondria and the levels of ROS significantly decreased. The significant advantage of a low-dose concentration is the high targeted protection of mitochondria by antioxidants. Due to the triphenylphosphonium cations linked with tempo, Mito-TEMPO is rapidly absorbed by mitochondria and eliminates overproduced ROS [14]. Furthermore, Mito-TEMPO improved human post-thaw sperm mitochondrial membrane potential [19]. In rooster sperm cooling processes, the addition of Mito-TEMPO to the storage medium can protect mitochondrial activity and decrease lipid peroxidation [20]. These results demonstrate that Mito-TEMPO significantly enhanced mitochondrial function, which is consistent with our observations. In addition, vitrification led to a significant decrease in ATP content in oocytes, and Mito-TEMPO was able to reverse this effect significantly. Evidence indicates that ATP production defects lead to meiotic oocyte errors [36]. Such errors caused incorrect first PB extrusion, irregular chromosome distribution, and aneuploidy. Previous studies reported similar results and found that heat production increased when ATP declined in mice vitrified oocytes [28]. Another piece of research proved that Mito-Q can increase ATP during porcine oocyte maturation [37].

Another important role of mitochondria in oocytes is the storage and transportation of calcium, which can have both beneficial and detrimental effects [38]. Under low Ca^2+^ conditions, ATP synthesis was reduced in oocytes [39]. Excessive Ca^2+^ may increase mitochondrial oxidative stress, ultimately resulting in irreversible cell death [40,41]. Recent research has indicated that the vitrification of immature oocytes leads to an increase in mitochondrial Ca^2+^ levels [42]. In our research, we also verified that Mito-TEMPO, as anticipated, improved the levels of mitochondrial and cytoplasmic Ca^2+^. The findings of our study suggest that Mito-TEMPO could help maintain Ca^2+^ homeostasis and ATP levels in vitrified oocytes through the enhancement of mitochondrial quality.

Single-cell transcriptome profiling was used to study the mechanisms by which Mito-TEMPO improves the quality of vitrified oocytes, aiming to identify potential target effectors. The results of our study indicated a mis-expression of genes linked to the mitochondrial respiratory chain and inner membrane in vitrified oocytes. Vitrification led to a decrease in *ATP8* and *ND4L* expression in sheep oocytes, which was subsequently restored by applying Mito-TEMPO. *ATP8* is a subunit of ATP synthase, while *ND4L* is a subunit of mitochondrial respiratory chain complex I. They play a role in mitochondrial oxidative phosphorylation and ATP synthesis [43,44]. The findings indicate that the mitochondrial respiratory chain may mediate the effect of Mito-TEMPO on vitrified oocytes. The results of our study support the idea that vitrification disrupts the mitochondrial activity of oocytes [45]. In contrast, the analysis of mitochondrial distribution, ROS, ∆Ψm, and ATP content demonstrated that Mito-TEMPO could reverse mitochondrial dysfunction induced by oocyte vitrification.

The equilibrium between oxidant and antioxidant is crucial, as any imbalance may lead to potential damage. Consequently, we added 1 μM Mito-TEMPO to the in vitro maturation medium of oocytes that were not subjected to vitrification to investigate the potential effects of Mito-TEMPO on oocytes. An increase in the PBE rate was also observed after supplementation with 1 μM Mito-TEMPO in oocytes without vitrification, and there was an improvement in mitochondrial function that is consistent with previous research findings in bovine and porcine [16,46]. The beneficial effects of Mito-TEMPO on mitochondrial function have been previously documented. The selection of testing concentrations, ranging from 0.1 to 100 μM, was informed by previous research conducted on other species and cell types. The addition of 0.1 μM Mito-TEMPO to the in vitro maturation medium of porcine oocytes showed beneficial effects [17]. Our findings are consistent with those of Yousefian et al., who also observed the positive effects of 1 μM Mito-TEMPO on the quality of bovine oocytes and their potential for fertilization [16]. In this study, we indicated the presence of a positive effect of Mito-TEMPO in the maturation medium for oocytes without vitrification. However, it is still unclear whether the addition of Mito-TEMPO to post-thawing IVM medium alone can achieve the same effect. Meanwhile, it is also necessary to clarify the impact of Mito-TEMPO on in vitro sheep embryo development in further investigations.

## 5. Conclusions

In conclusion, Mito-TEMPO enhances the meiotic competence and mitochondrial function of vitrified sheep GV oocytes by modulating the respiratory chain, which includes suppressing the accumulation of mitochondrial and cytoplasmic ROS, regulating mitochondrial ATP production, and maintaining mitochondrial and cytoplasmic calcium homeostasis.

## Figures and Tables

**Figure 1 animals-14-00152-f001:**
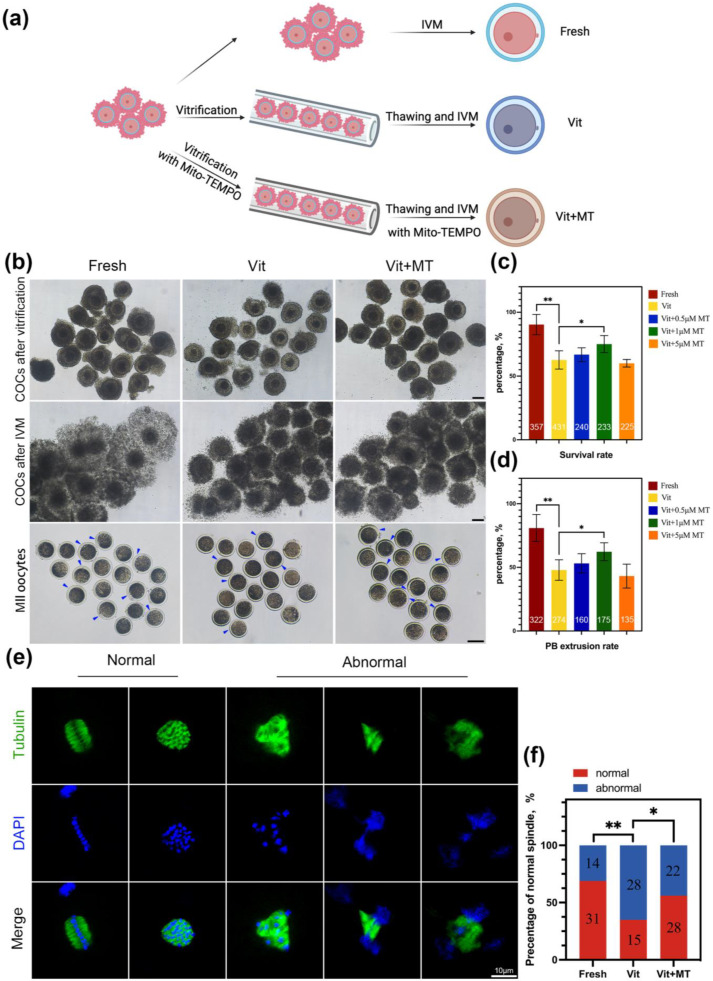
Effect of Mito-TEMPO on the survival and meiotic development of vitrified oocytes. (**a**) Schematic representation of vitrification-warming procedures for oocytes; (**b**) exemplary image of oocytes developed in vitro from fresh, vit, and vit + MT group oocytes. The blue represents PBE, scale bar = 100 μm; (**c**) post-IVM survival rate; (**d**) PBE rate. (**e**) The use of immunofluorescence to stain mature oocytes for α-tubulin (in green) and chromosome (in blue), scale bar = 10 μm; (**f**) comparison of normal spindle formation in the fresh (*n* = 45), vit (*n* = 43) and vit + MT (*n* = 50) groups. The number represents the number of oocytes used in this experiment. * *p* < 0.05, ** *p* < 0.01.

**Figure 2 animals-14-00152-f002:**
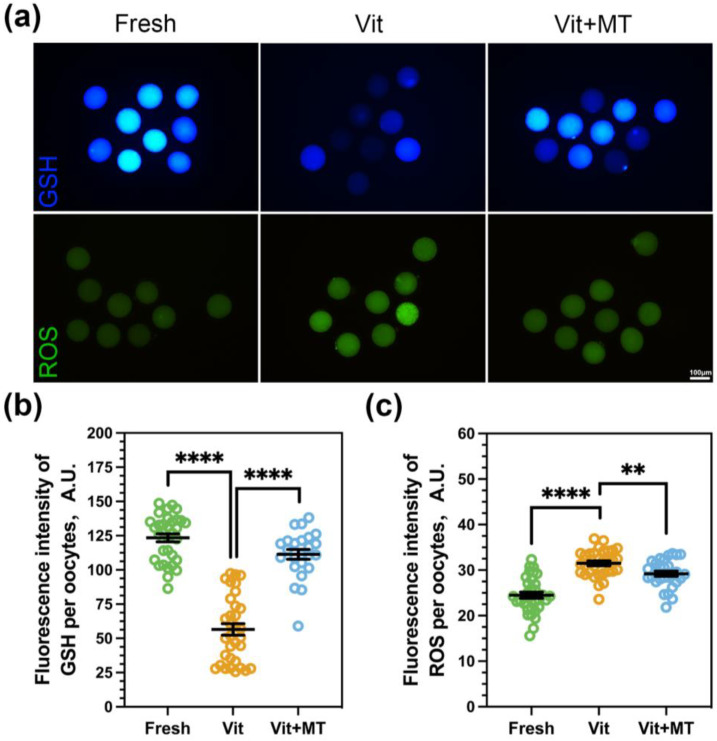
Effects of Mito-TEMPO on the redox status of vitrified oocytes. (**a**) Levels of ROS and GSH in oocytes across various groups, scale bar = 100 μm; (**b**) The fluorescence intensity of the GSH signal in the fresh (*n* = 33), vit (*n* = 33), and vit + MT (*n* = 24) groups; (**c**) the fluorescence intensity of the ROS signals in the fresh (*n* = 34), vit (*n* = 31), and vit + MT (*n* = 29) groups. ** *p* < 0.01, **** *p* < 0.0001.

**Figure 3 animals-14-00152-f003:**
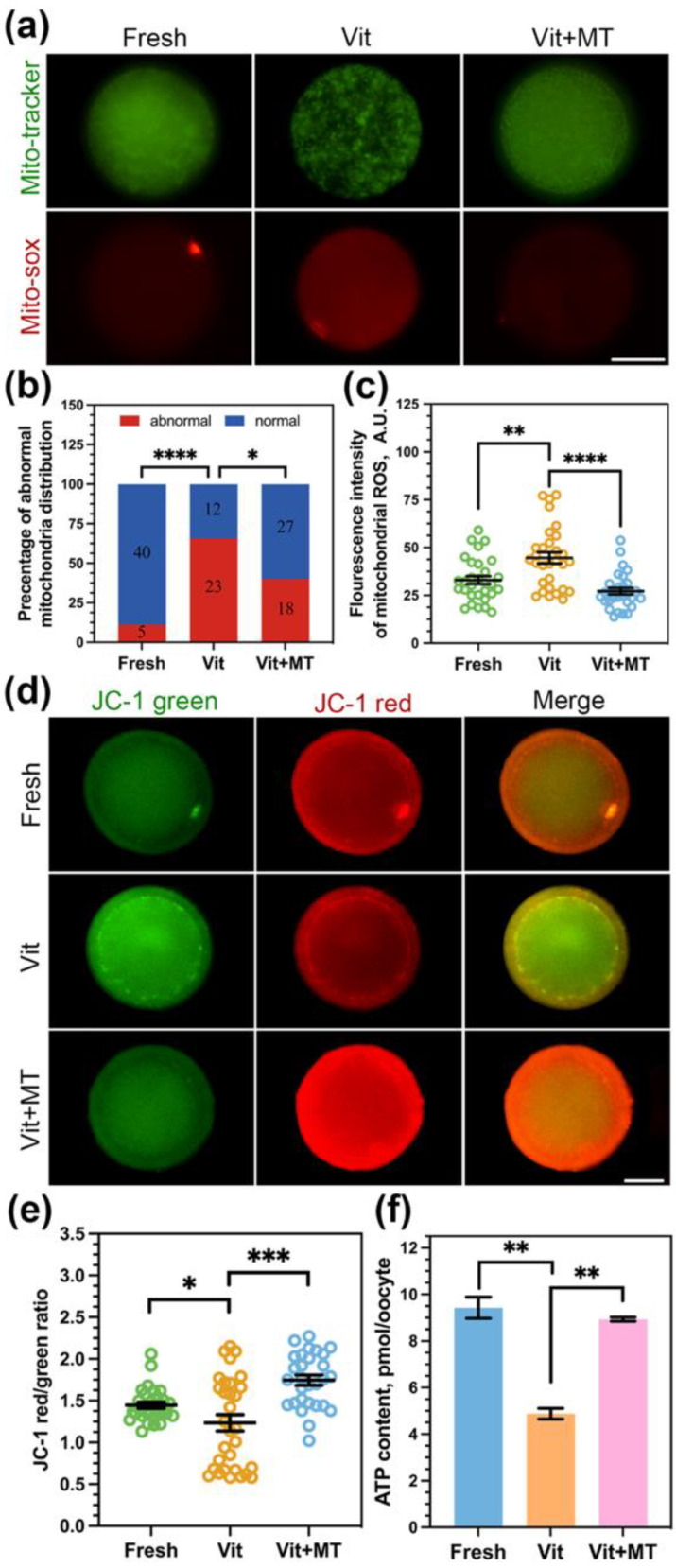
Effect of Mito-TEMPO on mitochondrial function in vitrified oocytes. (**a**) Representative images of mitochondria distribution and mitochondrial ROS levels in different groups; (**b**) rate of abnormal mitochondria distribution in fresh (*n* = 45), vit (*n* = 35), and vit + MT (*n* = 45) groups; the number represents the number of oocytes used in this experiment; (**c**) the fluorescence intensity of mitochondrial ROS signals in the fresh (*n* = 30), vit (*n* = 30), and vit + MT (*n* = 30) groups; (**d**) mitochondrial membrane potential was detected by JC-1 staining; (**e**) quantification of the mitochondria membrane potential level in the fresh (*n* = 30), vit (*n* = 31), and vit + MT (*n* = 29) groups; (**f**) oocytes ATP content was measured in the fresh (*n* = 40), vit (*n* = 40), and vit + MT (*n* = 40) groups. Scale bar = 50 μm. * *p* < 0.05, ** *p* < 0.01, *** *p* < 0.001, and **** *p* < 0.0001.

**Figure 4 animals-14-00152-f004:**
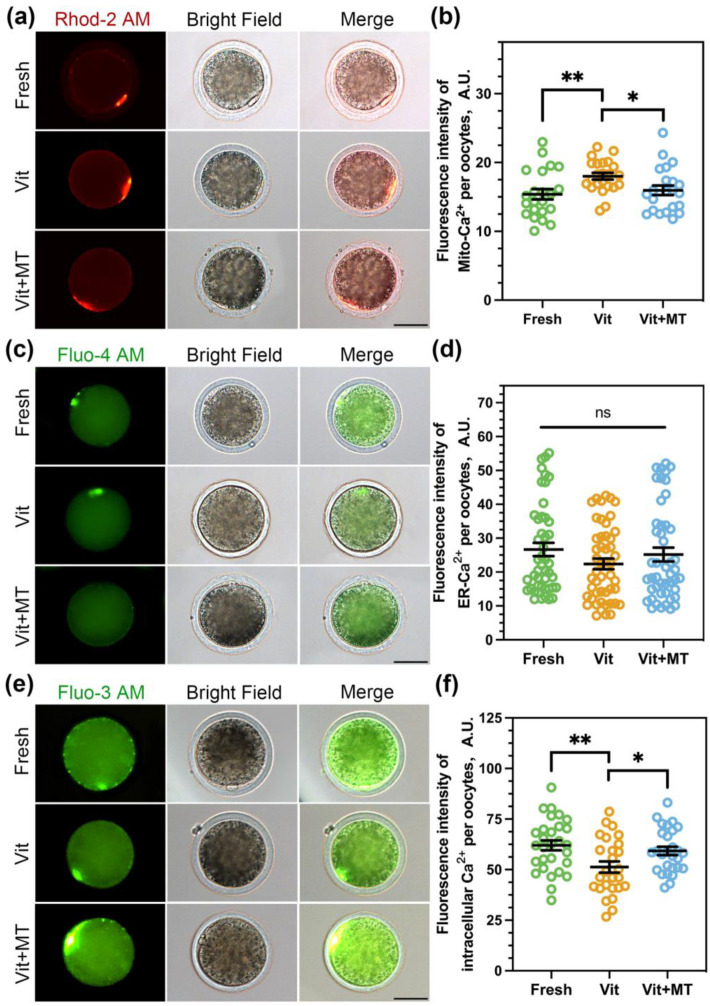
Effects of Mito-TEMPO on [Ca^2+^]_m_, [Ca^2+^]_ER_, and [Ca^2+^]_c_. (**a**) Representative images of [Ca^2+^]_m_ in different groups; (**b**) quantification of Rhod 2-AM fluorescence intensity in the fresh (*n* = 22), vit (*n* = 23), and vit + MT (*n* = 22) groups; (**c**) representative images of [Ca^2+^]_ER_ in different groups; (**d**) quantification of Fluo 4-AM fluorescence intensity in the fresh (*n* = 47), vit (*n* = 49), and vit + MT (*n* = 47) groups; (**e**) representative images of [Ca^2+^]_c_ in the different groups; (**f**) quantification of Fluo 3-AM fluorescence intensity in the fresh (*n* = 29), vit (*n* = 26), and vit + MT (*n* = 28) groups. Scale bar = 50 μm. * *p* < 0.05 and ** *p* < 0.01; ns represents no significance.

**Figure 5 animals-14-00152-f005:**
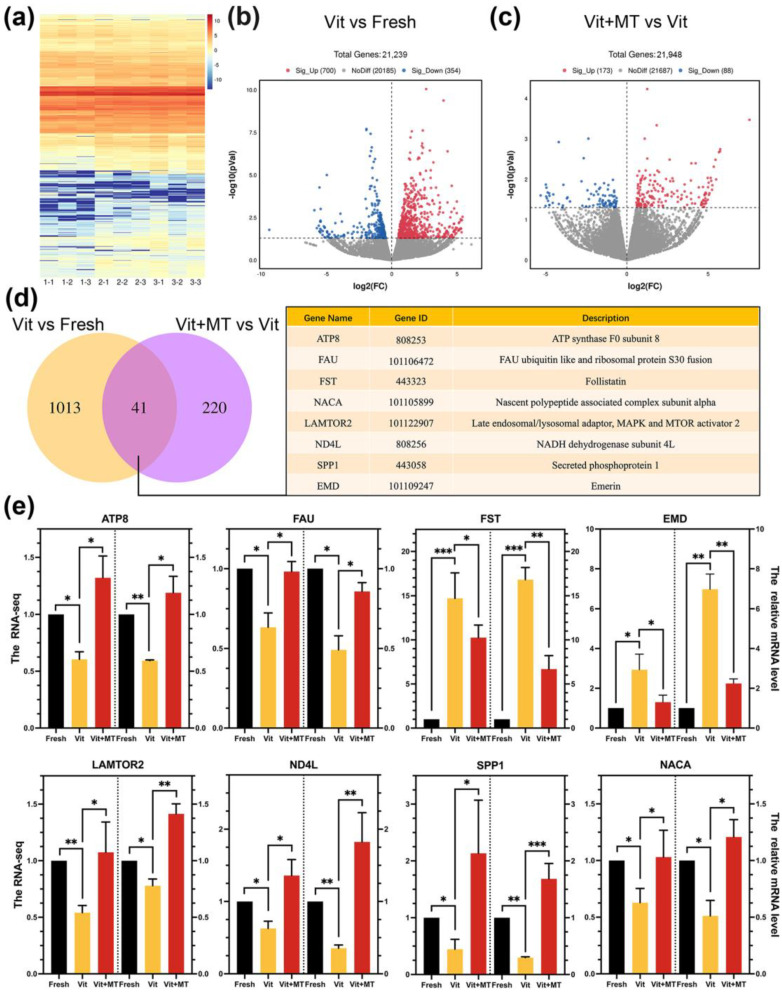
Impact of Mito-TEMPO on the transcriptome profiling of vitrified oocytes. (**a**) The heatmap illustrates the gene expression of the fresh (1-1, 1-2, and 1-3), vit (2-1, 2-2, and 2-3), and vit + MT (3-1, 3-2, and 3-3) groups oocytes; (**b**) the volcano plot shows DEGs (downregulated, blue; upregulated, red) in the vit group vs. the fresh group; (**c**) the volcano plot displays the DEGs (downregulated, blue; upregulated, red) in vit + MT group vs. the vit group; (**d**) the Venn diagram represents the common DEG distributions (some DEG information is listed); (**e**) the verification of expression the levels of the DEGs were verified by RT-qPCR; * *p* < 0.05, ** *p* < 0.01, and *** *p* <0.001.

**Figure 6 animals-14-00152-f006:**
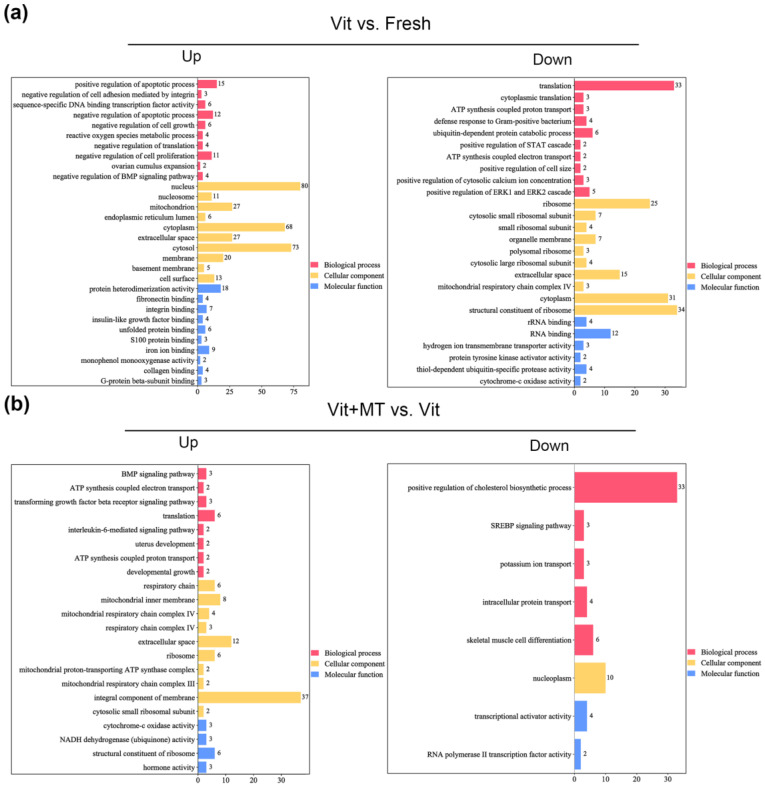
The GO enrichment analysis of DEGs in different groups. (**a**) Analysis of DEGs that were upregulated and downregulated in the vit group vs. the fresh group; (**b**) analysis of DEGs that were upregulated and downregulated in the vit + MT group vs. the vit group. Red represents biological processes, yellow represents cellular components, and blue represents molecular function.

**Figure 7 animals-14-00152-f007:**
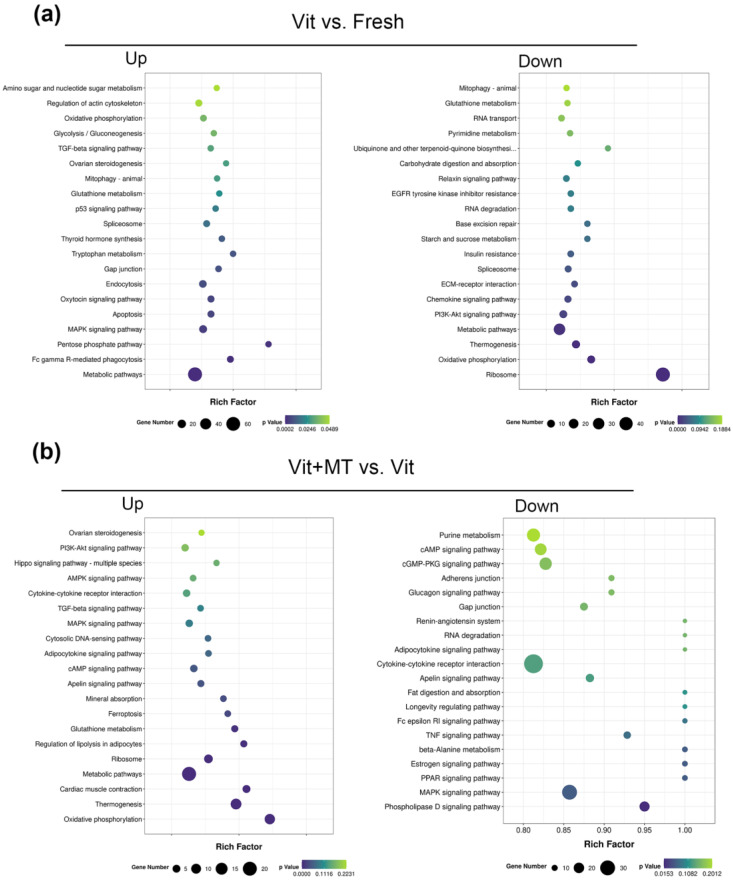
KEGG enrichment analysis of DEGs in different groups. (**a**) Analysis of DEGs that were upregulated and downregulated in the vit group vs. the fresh group; (**b**) analysis of DEGs that were upregulated and downregulated in the vit + MT group vs. the vit group.

**Figure 8 animals-14-00152-f008:**
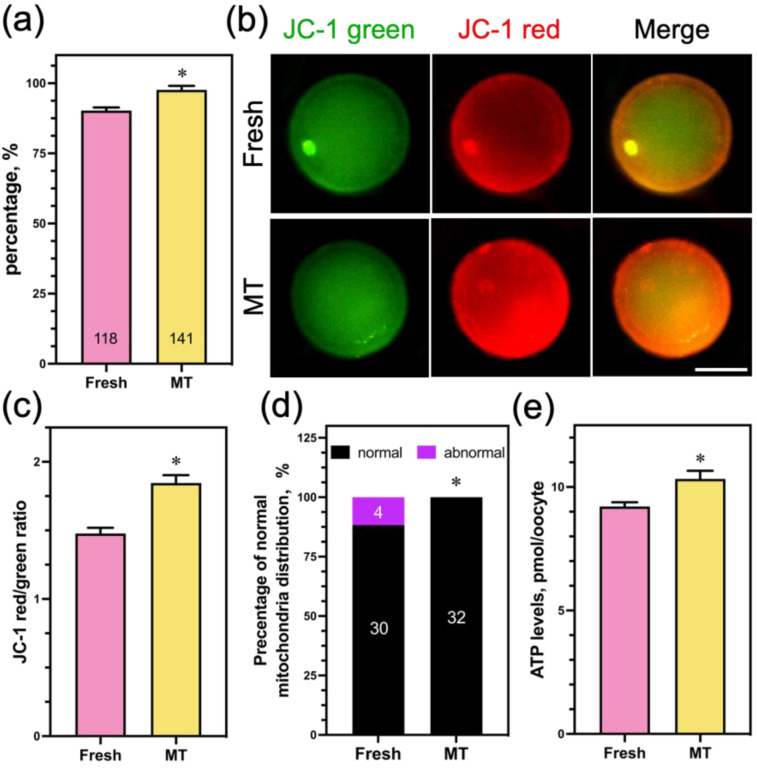
Effect of Mito-TEMPO on fresh oocyte mitochondrial function. (**a**) The rate of PB extrusion; (**b**) mitochondrial membrane potential was detected using JC-1 staining; (**c**) quantification of the mitochondria membrane potential level in the fresh (*n* = 25) and MT (*n* = 23) groups; (**d**) rate of normal mitochondria distribution in the fresh (*n* = 34) and MT (*n* = 32) groups; (**e**) oocytes ATP content was measured in the fresh (*n* = 4 0) and MT (*n* = 40) groups. Scale bar = 50 μm. The data are presented as the mean percentage (mean ± SEM) of at least three independent experiments; the number represents the number of oocytes used in this experiment; * *p* < 0.05.

**Table 1 animals-14-00152-t001:** Primer sequences used for RT-qPCR.

Gene	NCBI Reference Sequences	Primer Sequence (5′ to 3′)	Annealing Temperature (°C)
*GADPH*	NM_001190390	F: GGTTGTCTCCTGCGACTTCA R: CAGGGCCTTGAGGATGGAAA	55
*ATP8*	XR_003587932.1	F: GCCACAACTAGACACATCAACG R: AGGGGTAATGAAAGAGGCAAA	55
*FAU*	XM_004019665.4	F: TAGCTTCGTTGGAGGGCATC R: TTAGGGCCCTTCTTCTTGCC	55
*FST*	XM_012096672.3	F: TATGGGACTTCAAGGTTGGC R: GGTGTCTTCCGAAATGGAGT	50
*NACA*	XM_012174719.3	F: CAACAATGCCCTCTGGCAAC R: TTCTGAGCGAGCAACTGGAG	50
*LAMTOR2*	XM_004009674.3	F: AGACCGTTGGCTTCGGAATG R: TGATGCTGCTACTTGGGTGAG	55
*ND4L*	XR_003587932.1	F: TCACAGTATCCCTCACAGGACT R: CTCGCAAGCTGCGAAAACTA	55
*SPP1*	NM_001009224.1	F: TCCGCCCTTCCAGTTAAACC R: TCAGGGGTTTCAGCATCGTC	55
*EMD*	XM_027963104.1	F: CGCCAGTACAACATCCCACA R: ACGGACGCCGAATCTAAGTC	55

## Data Availability

The data presented in this study are available on request from the corresponding author.

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
