# Peer review of "Mito-TEMPO Improves the Meiosis Resumption and Mitochondrial Function of Vitrified Sheep Oocytes via the Recovery of Respiratory Chain Activity"

_animals, 2024, doi:10.3390/ani14010152_

Round 1

Reviewer 1 Report

Comments and Suggestions for Authors

Materials and Methods

1. Experimental groups are unclear:  line 106 - 4 groups are listed (M&M), while in the Results (lines 222-223; Fig. 1a; ) 3 groups are described.

If I guess correctly, two (or even 3 series of experiments) were carried out in the study: in the first series, the dose of Mito-Tempo (MT) was determined; in the second one, the selected dose (1µM) of MT was used for subsequent analyses, and additionally experiments presented on Fig. 6 (the third series) were performed.

The experimental design is needed. In each series of experiments, the experimental groups should be clearly described. 

Results

2. “survival rate” -  how was it calculated ? - it should be explained.  Meiotic maturation - whether the stages of meiosis (GV, GVBD, MI, MII, degenerated oocytes) were assessed or only the rate of BP extrusion was calculated? If so, Figure 1c - is redundant as the more important results have been presented on Figure 1d (PBE rate);   

3. Fig 1b (PB: third row)  – red arrow represent apoptotis – it should be explained why these cells were considered apoptotic

4. Fig 2 b, c: GSH/ROS pre oocyte – it should be: “per oocyte”

5. Figure 5 – too much data is presented in Fig. 5, it is difficult to read. I suggest presenting these results in at least in two separate Figures

 6. The most important: Three groups were compared in the study: 1. Fresh – IVM (Fresh); 2. Vit/thawing – and IVM (Vit) and 3. Vit+MT/thawing – and IVM+MT (Vit-MT) -  as shown in Fig 1a.

However, this experimental layout does not fully indicate to what extent COCs vitrification with MT itself had a protective effect on mitochondria, and to what extent this effect was influenced by the presence of MT during post-thawed IVM (or both simultaneously)?

As shown in Fig. 6. culture of Fresh-oocytes in medium supplemented with MT (IVM+MT) had a positive effect on PBE rate and mitochondria parameters, so perhaps it is enough to add MT to IVM of oocytes vitrified without MT, to achieve the same effect ?

To fully demonstrate the protective role of Mito-Tempo to mitochondria in response to vitrification, one more control experiment should be performed: vitrification without MT followed by IVM with MT after thawing (i.e. Vit/thawing -IVM+MT)

7. Disscussion:

375: delete “stimuli” please

The selected genes (listed in the table in Fig 5 d) should be discussed

Author Response

Response to Reviewer 1 comments

We feel great thanks for your professional review work on our article. As you concerned, there are several problems that need to be addressed. According to your nice suggestions, we have made extensive corrections to our previous draft, the detailed corrections are listed below.

Comments 1: The experimental design is needed. In each series of experiments, the experimental groups should be clearly described.

Response 1: We have added experimental design in lines 222-242 of the revised manuscript according to your suggestion.

Comments 2: “survival rate” - how was it calculated? - it should be explained. Meiotic maturation - whether the stages of meiosis (GV, GVBD, MI, MII, degenerated oocytes) were assessed or only the rate of BP extrusion was calculated? If so, Figure 1c - is redundant as the more important results have been presented on Figure 1d (PBE rate).

Response 2: The calculation method of survival rate was added in lines 128-133 of the revised manuscript. For meiotic maturation, only PBE rate was calculated. However, we did not remove survival rate, PBE rate is calculated based on the survival oocytes.

Comments 3: Fig 1b (PB: third row) – red arrow represent apoptotis – it should be explained why these cells were considered apoptotic

Response 3: We referred to previous methods to determine oocytes survival and apoptosis based on morphological evaluation. Undaring light microscope, the red arrow oocytes have faint cytoplasm and unclear membrane structure. We have added this reference to the revised manuscript in the lines 130-132.

Comments 4: Fig 2 b, c: GSH/ROS pre oocyte – it should be: “per oocyte”

Response 4: Thank you for pointing this out, and we have corrected the “pre” into “per”. The full text was checked and revised.

Comments 5: Figure 5 – too much data is presented in Fig. 5, it is difficult to read. I suggest presenting these results in at least in two separate Figures

Response 5: Thank you for pointing this out. To facilitate reading, we have divided Figure 5 into three separate figures presentations in revised manuscript.

Comments 6: To fully demonstrate the protective role of Mito-Tempo to mitochondria in response to vitrification, one more control experiment should be performed: vitrification without MT followed by IVM with MT after thawing (i.e. Vit/thawing -IVM+MT)

Response 6: We agree that more studies would be useful to understand the role of Mito-TEMPO to mitochondria in response to vitrification or in vitro maturation. At present, we have determined the protective effect of Mito-TEMPO on mitochondria in vitrified oocytes by referring to previous studies of both vitrified-thawing and in vitro maturation with addition of antioxidants in vitrification of GV stage oocytes. We will further improve this part in future experiments to compare the effects of Mito-TEMPO added in vitrification or in vitro maturation processes.

Comments 7: L.375: delete “stimuli” please; The selected genes (listed in the table in Fig 5 d) should be discussed

Response 7: We deleted the word “stimuli”. And added a description of the validation genes to the discussion, which is in lines 393-395 of revised manuscript.

Reviewer 2 Report

Comments and Suggestions for Authors

Although the manuscript presented for review definitely does have a scientific value and provides novel, previously unpublished data, it has several drawbacks. Generally, the method description needs to be improved, the experimental design needs to be outlined clearly, graphical presentation of results must be amended sometimes. In addition, major English corrections are required, since wrong wording and inappropriate for a research publication writing style distract the reader from the essence of the study. Some mistakes are directly addressed in my comments, but not all – a thorough revision is needed.

Introduction

L.1 The title sounds confusing. A rewording is necessary (e.g. smth. like ‘Mito-TEMPO Improves Meiosis Resumption and Mitochondrial Function by Respiratory Chain in Vitrified Sheep Oocytes’?)

L.107 The selected COCs were randomly divided into four groups: control, vitrification, and Mito-TEMPO with and without vitrified groups. – Must be rewritten.

L.127 Section 2.4. – Where is the secondary Ab step for alfa-tubulin?

L139-140 – Why are the concentrations written in words?

M&Ms

Experimental design is not clearly described.  When exactly was MitoTempo added for vitrification and thawing/ culture? The study groups are poorly described in M&Ms section. There should be a clear outline of the design, with MitoTempo concentrations and with numbers of oocytes in each group.

From how many animals were the ovaries collected for each experiment? Were the oocytes from different animals pooled together and then randomly allocated between experimental groups?

Fluorescence: how were the images of fluorescence for later quantitative analysis taken to ensure that the differences in the signal are real and not the result of, e.g. automated correction of the exposure time?

What was the reason to use both MitoTRACKER and JC-1, when they both show at the end the same thing - active mitochondria?

Figure 1. How was the survival assessed? (include in M&Ms)

Figure 1 (b) When were the pictures of GV oocytes taken, right after vitrification? If after vitrification, then a representative photo of the morphology prior to vitrification has to be included, otherwise it looks like the vitrification group had poorer cumulus from the beginning.

Figure 1 (c), (d) – Numbers on the bars are barely visible even at a large magnification, maybe remove ‘n=’ leaving just the numbers?

Figure 4. Typing mistake – Flou instead of Fluo

Figure 4 a – I don’t see any difference in the signal between fresh and Vit+MT

Figure 5 f-I – The graphs are not readable. In this format they make no sense, need to be either enlarged or removed with detailed description in the text.

On Figure 6d the oocyte from the MitoTEMPO group is significantly larger than the control one. Was the intensity of the fluorescent signal calculated per unit of square or per the oocyte?

Discussion

L.376-377 – re-word

L.377 – why are GV oocytes so sensitive to cryoprocedures? In general or in sheep? Why? References are needed.

General comment:

Study group names are not kept consistent throughout the publication. Should be consistent with the figures as well, so that also in the text they are titles as Fresh Group (or CTRL Group), Vitrification Group, Vitrification+MT Group or similarly.

Comments on the Quality of English Language

As I have written above, English requires significant revision. 

Author Response

Response to Reviewer 2 comments

We feel great thanks for your professional review work on our article. As you concerned, there are several problems that need to be addressed. According to your nice suggestions, we have made extensive corrections to our previous draft, the detailed corrections are listed below.

Comments 1: L.1 The title sounds confusing. A rewording is necessary (e.g. smth. like ‘Mito-TEMPO Improves Meiosis Resumption and Mitochondrial Function by Respiratory Chain in Vitrified Sheep Oocytes’?)

Response 1: In the revised manuscript, we reword title in to “Mito-TEMPO Improves Meiosis Resumption and Mitochondrial Function of Sheep Vitrified Oocyte by Recover Respiratory Chain Activity”.

Comments 2: L.107 The selected COCs were randomly divided into four groups: control, vitrification, and Mito-TEMPO with and without vitrified groups. – Must be rewritten.

Response 2: We have added the experimental design and supplemented the detailed grouping in lines 223-226 of revised manuscript.

Comments 3: L.127 Section 2.4. – Where is the secondary Ab step for alfa-tubulin?

Response 3: We purchased anti-?-tubulin-FITC antibody (F2168) at sigma. This antibody does not require a secondary antibody when used. In the revised manuscript, we changed the previously misleading description in line 138 of the materials and methods.

Comments 4: L139-140 – Why are the concentrations written in words?

Response 4: Thank you for pointing this out, and we have corrected the concentration into number. The full text was checked and revised.

Comments 5: Experimental design is not clearly described.  When exactly was MitoTempo added for vitrification and thawing/ culture? The study groups are poorly described in M&Ms section. There should be a clear outline of the design, with MitoTempo concentrations and with numbers of oocytes in each group.

Response 5: We have added the experimental design in line 222-242 of the revised manuscript, divided the experiment into three sections, and described the Mito-TEMPO addition time in each section.

Comments 6: From how many animals were the ovaries collected for each experiment? Were the oocytes from different animals pooled together and then randomly allocated between experimental groups?

Response 6: The experiment was done several times, and we are sorry that we did not count the number of animals in each experiment. We collected oocytes from different animals and pooled together, and then selected COCs for each group at random. We have added this description in both oocyte collection and experimental design parts in revised manuscript.

Comments 7: Fluorescence: how were the images of fluorescence for later quantitative analysis taken to ensure that the differences in the signal are real and not the result of, e.g. automated correction of the exposure time?

Response 7: Fluorescence intensity was quantified under the same staining procedure and parameters through confocal laser scanning microscope, and we have added this description in materials and methods parts of the revised manuscript.

Comments 8: What was the reason to use both MitoTRACKER and JC-1, when they both show at the end the same thing - active mitochondria?

Response 8: During oocyte maturation, mitochondrial distribution changes markedly. Previous studies have shown that the quality of oocyte maturation in vitro was associated with changes in distribution of active mitochondria. Meanwhile, the stain of Mito-tracker is not dependent upon mitochondrial membrane potential. So, we wanted to elucidate the beneficial effects of Mito-TEMPO on mitochondrial function by using both Mito-tracker and JC-1.

Comments 9: Figure 1. How was the survival assessed? (include in M&Ms)

Response 9: We referred to previous methods to determine oocytes survival and apoptosis based on morphological evaluation. Undaring light microscope, the red arrow oocytes have faint cytoplasm and unclear membrane structure. We have added this reference to the revised manuscript in the lines 130-132.

Comments 10: Figure 1 (b) When were the pictures of GV oocytes taken, right after vitrification? If after vitrification, then a representative photo of the morphology prior to vitrification has to be included, otherwise it looks like the vitrification group had poorer cumulus from the beginning.

Response 10: We have changed the caption in Figure 1b to show the time of shooting. And we are sorry that we didn't put the representative photo of the morphology prior to vitrification because we didn’t take it before vitrification. We hope that the additional descriptions in line 109 of the materials and methods will allow the reader to understand that all oocytes are randomly grouped.

Comments 11: Figure 1 (c), (d) – Numbers on the bars are barely visible even at a large magnification, maybe remove ‘n=’ leaving just the numbers?

Response 11: Thanks for your suggestion. We have removed “n=” and enlarged number for read easily in the revised manuscript.

Comments 12: Figure 4. Typing mistake – Flou instead of Fluo.

Response 12: Thank you for pointing this out, and we have corrected the “Flou” into “Fluo” in the revised manuscript. The full text was checked and revised.

Comments 13: Figure 4 a – I don’t see any difference in the signal between fresh and Vit+MT.

Response 13: Thank you very much for your careful reading. The fluorescence intensity statistics also showed that no significant difference between fresh and vit+MT groups.

Comments 14: Figure 5 f-I – The graphs are not readable. In this format they make no sense, need to be either enlarged or removed with detailed description in the text.

Response 14: Thank you for pointing this out. To facilitate reading, we have divided Figure 5 into three separate figures presentations in revised manuscript.

Comments 15: On Figure 6d the oocyte from the MitoTEMPO group is significantly larger than the control one. Was the intensity of the fluorescent signal calculated per unit of square or per the oocyte?

Response 15: In this experiment, we measured the mean fluorescence intensity of per oocyte. And we have changed the previously misleading description in materials and methods of the revised manuscript in line 168-169.

Comments 16: L.376-377 – re-word

Response 16: We reword lines 376-377 as follows. “The most immediate effect of vitrification on GV stage oocytes is a decrease in germinal vesicle breakdown and polar body extrusion. Furthermore, GV stage oocytes are more susceptible to damage during vitrification compared to other developmental stages.”

Comments 17: L.377 – why are GV oocytes so sensitive to cryoprocedures? In general or in sheep? Why? References are needed.

Response 17: We sincerely appreciate the valuable comments. Several studies revealed that MII oocytes had better resistance to cryopreservation than immature oocytes both in pig and in cattle. Mo had shown that GV stage oocytes had lower developmental competence after vitrification compared with GVBD, Mâ…  and Mâ…¡ stages, and we have added this reference to the revised manuscript in line 399.

Comments 18: Study group names are not kept consistent throughout the publication. Should be consistent with the figures as well, so that also in the text they are titles as Fresh Group (or CTRL Group), Vitrification Group, Vitrification+MT Group or similarly.

Response 18: We checked the full text to unify the descriptions of the experimental groups in fresh group, vit group, vit+MT group and MT group in revised manuscript.

Response to Comments on the Quality of English Language

Point 1: As I have written above, English requires significant revision. 

Response 1: Regarding the comments on the language, we invited a friend of us who is a native English speaker form USA to help polish our article. We hope the revised manuscript could be acceptable for you.

Round 2

Reviewer 1 Report

Comments and Suggestions for Authors

Overall, most of my comments have been taken into account, but the manuscript still requires some corrections.

1. First of all, the experimental design (lines 223-242) are still unclear. The groups, i.e when exactly (for vitrification or/and for post-thawing IVM) MT was added, must be clearly defined in this section, not only visualized in the Fig.1a; Lines 223-226 - must be reworded or even deleted, and a detailed description of the three groups can be given e.g., after the determining the optimal dose of MT (Experiment 1).

 2. Experiment 3: The wording “toxic effect” is awkward/inappropriate, replace with e.g. MT improve mitochondrial function. The lack of the toxic effect of MT on oocyte survival and PBE rate (thus indirectly on mitochondrial function) was already demonstrated in Experiment 1, when the optimal dose of MT was determined. Experiment 3 directly demonstrated the positive effect of MT on meiotic maturation and mitochondrial function of fresh (non-vitrified) oocytes. However, in my opinion, the group Fresh/IVM+MT should have already been planned and implemented in experiment 1, and not as an additional experiment 3. The same applies to the missing Vit-thawing/IVM+MT group.

3. Line128: 2.4 Survival and the first polar body extrusion (PBE) –  “Oocyte” is missing:   “Oocyte survival and  ..  “; The description of survival rate evaluation should be reworded.

 4. “Identical staining method” (lines: 149, 167,188);  “level of oocyte fluorescence” (169)   - should be clarified

 5. Line 253: Different concentrations of Mito-TEMPO (0, 0.5, 1, 5 μM) were applied  during vitrified-thawed and in vitro development.  -  „and” - suggests that MT was also added to IVM after thawing - is it so?

 6. Line 373, 380 ...” MT toxic effect”  - should be reworded, see comment 2

 7. Fig.1 line 267 – „apoptotis” - I suggest caution in assessing apoptosis based on the light appearance of the ooplasm and unclear membrane only. The color/degree of "darkness" of the ooplasm is related to the intracytoplasmic lipid content and a light apparence of the ooplasm does not indicate “not viable” oocyte. Moreover, in the paper referred to by the authors, the viability of the oocyte (or rather its lack), was validated with PI staining, which is usually used to identify dead cells. No such analysis was carried out in the reviewed study, so the authors can only suspect that the cells were dead. To assess apoptosis in cells, additional staining with apoptosis-specific detection kit is required, so I my opinion the red arrows and “apoptosis” must be removed from Fig. 1.

8. Fig. 3d and 8b - JC-1 staining -  in the pictures we can see bright spots that look like nuclear chromatin - please clarify, how is this possible, since this dye does not make nuclear chromatin visible?   I have stained oocytes with JC-1 and after staining nuclear chromatin was never visible.

 9. Line 392:  I suggest “ meiotic maturation”  - instead  “meiosis”

10. Lines 402-403 and 451 – describe the same – MT increase PBE rate ..  – the text should not be repeated. Also remove all references to Figures from the Discussion, they are unnecessary.

General Comment, although the Vit-thawing/IVM+MT group is still missing, the authors demonstrated a positive effect of Mito-Tempo supplementation on mitochondrial functions and PBE of fresh and vitrified sheep oocytes. However, whether vitrification with Mito-Tempo is necessary or whether it is enough to add Mito-Tempo to post-thawing IVM to achieve the same effect it is still unclear - this issue should be at least discussed. 

Author Response

Response to Reviewer 1 comments

Thanks for your constructive comments and suggestions. We have substantially revised our manuscript after reading the comments. In the revised manuscript, all the changes are highlighted in the yellow and green for easy inspection. We hope this revision can make our paper more acceptable. The revisions were addressed point by point below.

Comments 1: First of all, the experimental design (lines 223-242) are still unclear. The groups, i.e when exactly (for vitrification or/and for post-thawing IVM) MT was added, must be clearly defined in this section, not only visualized in the Fig.1a; Lines 223-226 - must be reworded or even deleted, and a detailed description of the three groups can be given e.g., after the determining the optimal dose of MT (Experiment 1).

Response 1: Thanks for your suggestion. We have deleted lines 223-226 and reworded the detailed description in each experiment. We have moved the experiment design part in the beginning of materials and methods, which is in lines 90-113 of the revise manuscript.

Comments 2: Experiment 3: The wording “toxic effect” is awkward/inappropriate, replace with e.g. MT improve mitochondrial function. The lack of the toxic effect of MT on oocyte survival and PBE rate (thus indirectly on mitochondrial function) was already demonstrated in Experiment 1, when the optimal dose of MT was determined. Experiment 3 directly demonstrated the positive effect of MT on meiotic maturation and mitochondrial function of fresh (non-vitrified) oocytes. However, in my opinion, the group Fresh/IVM+MT should have already been planned and implemented in experiment 1, and not as an additional experiment 3. The same applies to the missing Vit-thawing/IVM+MT group.

Response 2: Thank you for pointing this out. We have replaced “toxic effect” with “potential effect” in line 107. In previous comments, another reviewer suggested that supplementing Mito-TEMPO in fresh oocytes individually to confirm the potential effect of optimal concentration of Mito-TEMPO on oocytes. They thought that the dose-dependent effect of Mito-TEMPO on vitrified oocytes was clearly a prelude of potential detrimental effect also in fresh oocytes.

Comments 3: Line128: 2.4 Survival and the first polar body extrusion (PBE) –  “Oocyte” is missing:   “Oocyte survival and  ..  “; The description of survival rate evaluation should be reworded.

Response 3: Thanks for your suggestion. We have reworded the title of 2.5 in line 142 and redescribed the survival rate evaluation in the lines 145-146.

Comments 4: “Identical staining method” (lines: 149, 167,188);  “level of oocyte fluorescence” (169)   - should be clarified

Response 4: Thank you for pointing this out, and we have corrected the description in the lines 162, 174, 180, 181 and 200.

Comments 5: Line 253: Different concentrations of Mito-TEMPO (0, 0.5, 1, 5 μM) were applied during vitrified-thawed and in vitro development.  -  „and” - suggests that MT was also added to IVM after thawing - is it so?

Response 5: We have reworded the description of this part in lines 243-244. We hope the reworded description will allow the reader to understand easily that different concentrations of Mito-TEMPO were added to both vitrification, thawing and in vitro maturation media.

Comments 6: Line 373, 380 ...” MT toxic effect” - should be reworded, see comment 2

Response 6: Thanks for your suggestion. We have reworded this fragment in lines 363 and 371.

Comments 7: Fig.1 line 267 – „apoptotis” - I suggest caution in assessing apoptosis based on the light appearance of the ooplasm and unclear membrane only. The color/degree of "darkness" of the ooplasm is related to the intracytoplasmic lipid content and a light apparence of the ooplasm does not indicate “not viable” oocyte. Moreover, in the paper referred to by the authors, the viability of the oocyte (or rather its lack), was validated with PI staining, which is usually used to identify dead cells. No such analysis was carried out in the reviewed study, so the authors can only suspect that the cells were dead. To assess apoptosis in cells, additional staining with apoptosis-specific detection kit is required, so I my opinion the red arrows and “apoptosis” must be removed from Fig. 1.

Response 7: Thanks for your suggestion. We have removed the red arrows in the Figure 1b. Morphological methods have been used to determine apoptosis in other studies, so we referred to the results of previous studies to retain the survival rate.

Comments 8: Fig. 3d and 8b - JC-1 staining - in the pictures we can see bright spots that look like nuclear chromatin - please clarify, how is this possible, since this dye does not make nuclear chromatin visible?   I have stained oocytes with JC-1 and after staining nuclear chromatin was never visible.

Response 8: Thanks for your careful reading. During oocyte maturation, the distribution of mitochondria changes significantly. After GVB and up to the MII stage, mitochondria are distributed equally in the ooplasm. As cytokinesis during meiosis is unequal, most mitochondria are moved back to the spindle poles and excluded from the cytoplasm along with the first polar body. So, we can see bright spots that look like nuclear chromatin when using JC-1. This same phenomenon can be observed in the results of other studies.

Comments 9: Line 392:  I suggest “ meiotic maturation”  - instead  “meiosis”

Response 9: Thanks for your suggestion. We have replaced “meiosis” with “meiotic maturation” in line 383.

Comments 10: Lines 402-403 and 451 – describe the same – MT increase PBE rate ..  – the text should not be repeated. Also remove all references to Figures from the Discussion, they are unnecessary.

Response 10: Thanks for your suggestion. We have removed all references to figures from the discussion, and redescribed lines 443-444 to distinguish from the preceding paragraphs.

Comments 11: General Comment, although the Vit-thawing/IVM+MT group is still missing, the authors demonstrated a positive effect of Mito-Tempo supplementation on mitochondrial functions and PBE of fresh and vitrified sheep oocytes. However, whether vitrification with Mito-Tempo is necessary or whether it is enough to add Mito-Tempo to post-thawing IVM to achieve the same effect it is still unclear - this issue should be at least discussed.

Response 11: Thanks for your suggestion. We have added the discuss in lines 454-457.

Reviewer 2 Report

Comments and Suggestions for Authors

In general, the authors have addressed most of my comments and the quality of presentation has improved. However, there are still some open questions that definitely must be addressed to avoid confusions by the readers.

Comments on the Quality of English Language

The style and grammar of the publication must be further improved, as in its current state the manusript does not conform to the high standards of academic writing. There are sometimes missing words, inappropriate word choice and confusing sentence structures. I am trying to address most of this in my comments.

Author Response

Response to Reviewer 2 comments

Thanks for your constructive comments and suggestions. We have substantially revised our manuscript after reading the comments. In the revised manuscript, all the changes are highlighted in the blue and green for easy inspection. We hope this revision can make our paper more acceptable. The revisions were addressed point by point below.

Comments 1: L.23 plentiful -abundant

Response 1: Thanks for your suggestion. We have replaced “plentiful” with “abundant” in line 23.

Comments 2: L.34 What is more – In addition

Response 2: Thanks for your suggestion. We have replaced “What is more” with “In addition” in line 34.

Comments 3: L.41 conserving - preserving

Response 3: Thanks for your suggestion. We have replaced “conserving” with “preserving” in line 41.

Comments 4: L.56 to minimize ice nucleation and growth - 56 to minimize ice nucleation and crystal growth

Response 4: Thanks for your suggestion. We have added “crystal” before “growth” in line 56.

Comments 5: L.65 Prior research has demonstrated that oocytes can directly utilize glucose with low efficiency [11, 12]. – This sentence is poorly connected to the context. What does it tell us?

Response 5: Thank you for pointing this out. In the earlier manuscript, the CCs results were included, and we removed them later for logical integrity. We have deleted this part in the beginning of second phases.

Comments 6: L.82 recovered oxidative stress and damage – decreased oxidative stress and damage.

Response 6: Thanks for your suggestion. We have replaced “recovered” with “decreased” in line 72.

Comments 7: L. 109 The selected COCs for each group at random -- ?

Response 7: Thank you for pointing this out. We have deleted this fragment in line 123 and reworded the description in experiment design.

Comments 8: L.130-133 Very bad writing, this fragment must be rewritten! And PB is EXTRUDED, not excreted

Response 8: Thank you for pointing this out, we have replaced “excreted” with “extrusion” in lines 146 and redescribed the line 143-147.

Comments 9: L.138 oocytes were reacted with an antibody (anti-?-tubulin, Sigma, USA, 1:2000) overnight at 4°C. ---- add the missing information, same as in the answers to the comments – _that the antibody was FITC conjugated

Response 9: Thank you for pointing this out, we have added the missing information of antibody in line 152.

Comments 10: L.139 - Then, we washed the oocytes thoroughly by soaking the buffer --- what?

Response 10: Thank you for pointing this out, we have reworded the description of this fragment in line 153.

Comments 11: L.148 - As a second step, the oocytes underwent a comprehensive cleansing process using PBS on no less than three times. – Replace with “After that, the oocytes were thoroughly washed with PBS at least three times”

Response 11: Thanks for your suggestion. We have replaced “As a second step, the oocytes underwent a comprehensive cleansing process using PBS on no less than three times” with “After that, the oocytes were thoroughly washed with PBS at least three times” in lines 161-162.

Comments 12: L.154 - were put – replace with ‚were placed‘

Response 12: Thanks for your suggestion. We have replaced “were put” with “were placed” in lines 159 and 178.

Comments 13: L.149 - Utilizing identical staining methods and settings --- ‘identical’ to what? In academic writing replace the word ‘identical’ to “the same”. Better add the washing and DAPI stating step, otherwise it is too confusing. L. 154 – cultured – replace with ‚incubated

Response 13: Thanks for your suggestion. We have replaced “identical” with “the same” in lines 162, 174, 180 and 200, and the word “cultured” replaced to “incubated” in line 167. In our experiments, except for the Immunofluorescence (IF) Staining part, we did not use the DAPI staining step.

Comments 14: L. 160 For MitoSOX, mean fluorescence intensity of per oocyte was quantified by using the same staining procedure and parameters. – the same for each oocyte? It is still confusing – was the exposure time always the same to exclude automated adjustment of the fluorescent image?

Response 14: Signals from oocytes were obtained using the same immunostaining procedure and setting up the same confocal microscope parameters (including exposure time, gain and intensity value). Then, using the ImageJ to define a region of each oocyte, and the mean intensity of fluorescence per unit area within the oocyte region was determined. The average intensity of all oocytes was used to compare the final mean fluorescence intensities between each group. We also reworded the description in lines 162, 174, 180 and 200, and hope it will be more easily for readers to understand.

Comments 15: L.166 the new description is still confusing

Response 15: In lines 180-181, we have replaced the description as below, “using the same confocal microscope parameters via a laser scanning confocal microscope”.

Comments 16: L. 168 The mean fluorescence intensity per unit area was assessed within the targeted area for each level of oocyte fluorescence – What do the authors mean by “for each level of oocyte fluorescence”?

Response 16: We have referred to the method described in previous study and replaced the description in lines 181-182 as below, “Mean fluorescence intensity of each oocyte was quantified using ImageJ software”.

Comments 17: L.169 utilizing the ImageJ software – change to ‘using ImageJ software’

Response 17: Thanks for your suggestion. We have replaced “utilizing the ImageJ software” with “using ImageJ software” in line 181.

Comments 18: L.174 – were quantified – replace with ‘was quantified’

Response 18: Thanks for your suggestion. We have replaced “were” with “was” in line 186.

Comments 19: L.176 blended through vertexing until lysis ----- replace with ‘vortexed until cells are fully lysed’

Response 19: Thanks for your suggestion. We have replaced “blended through vertexing until lysis” with “vortexed until the cells were fully lysed” in line 189.

Comments 20: L.188 – again ‘identical’…

Response 20: Thanks for your suggestion. We have replaced “identical” with “same” in line 200.

Comments 21: L.193 - Concisely, within the lysis buffer, each group had three specimens gathered (10 oocytes in each sample). – Do you mean that for each study group, 3 pooled samples were collected and each of these samples consisted of 10 oocytes pooled together?

Response 21: Yes. To reduce the error of single cell, we have increased the number of oocytes per sample to 10.

Comments 22: L.222 The Experimental Design section should be moved to the beginning of M&Ms. The Comment ( 5) regarding the number of the oocyte in each study group was not addressed.

Response 22: Thanks for your suggestion. We have moved the experimental design part to the beginning of M&Ms and added the number of the oocyte in each experiment.

Comments 23: L.223-226 – must be thoroughly rewritten.

Response 23: Thanks for your suggestion. We have deleted the previous lines 223-226 and reworded the detailed description in each Experiment.

Comments 24: L.228 - In this part, we added different concentrations of Mito-TEMPO (0, 0.5, 1, 5 μM) into the vitrified-thawing process and subsequent in vitro maturation ---- Rewrite as: ‘In this part, we added different concentrations of Mito-TEMPO (0, 0.5, 1, 5 μM) into vitrification, thawing and IVM media’

Response 24: Thanks for your suggestion. We have reworded the description of this fragment in lines 94-96.

Comments 25: L.233 What MI group was not included in the transcriptomic analysis?

Response 25: Do you mean that the MT group was not included in the transcriptomic analysis? If my conjecture is correct, we would like to focus primarily of this paper on the beneficial effects of Mito-TEMPO on vitrified oocytes, and we think the transcriptome analysis of fresh, vit, and vit+MT three groups is sufficient to explain the effect and target of Mito-TEMPO in vitrified oocyte.

Comments 26: Figure 1b – COCs before IVM – for the two vitrified groups are the images taken before or after vitrification?

Response 26: Thank you for pointing this out. We have replaced “COCs before IVM” with “COCs after vitrification” in figure 1b.

Comments 27: Comment 15 of the previous Comments – the mean fluorescence was measured per oocyte – what is your substantiation of this choice? Why in some cases you chose to measure per unit of square and in others – per oocyte despite the existing size differences?

Response 27: Thank you for pointing this out. We have unified the description and the full text has been checked and revised.

Comments 28: L.397-398 – the reference you chose [35] does not show any direct comparison of vitrification damage between different oocyte stages, and especially not in sheep. Rewrite this fragment to address the cause of higher sensitivity of GV stage oocytes and chose appropriate references.

Response 28: Thanks for your careful reading. In the previous [35] reference, the author collected sheep oocytes at various developmental stages including GV, GVBD, metaphase I and metaphase II and vitrified using OPS method. Survival and subsequent developmental rates were also assessed. Thus, we did not change the reference. In the revised manuscript, the serial number is [30].

Comments 29: L.412 the reference to Figures in the Discussion is not appropriate, this was already presented in the Results section.

Response 29: Thank you for pointing this out. We have removed all references to figures from the discussion.

Comments 30: L.416 – why ‘would’? It was clearly shown by other authors, so it must be stated this way.

Response 30: Thanks for your suggestion. We have reworded the description of this fragment in lines 406-409.

Comments 31: L.448 – ‘oxidant and antioxidant forces’ sounds awkward

Response 31: Thanks for your suggestion. We have reworded the description of this fragment in line 440.

Comments 32: L.454 Mito-TEMP – O is missing

Response 32: Thanks for your reminder and we have corrected the “Mito-TEMP” into “Mito-TEMPO” in line 446.

Comments 33: L.458 - yielded significant results --- reword, please

Response 33: Thank you for pointing this out and We have reworded this fragment in line 450.

Comments 34: L. 463Meiotic ability - meiotic competence

Response 34: Thanks for your suggestion. We have replaced “meiotic ability” with “meiotic competence” in line 459.

Comments 35: In general, throughout the Discussion section, species must always be indicated, when citing other’s works.

Response 35: Thank you for pointing this out. We have checked the discussion section and supplemented the species.

Response to Comments on the Quality of English Language

Point 1: The style and grammar of the publication must be further improved, as in its current state the manuscript does not conform to the high standards of academic writing. There are sometimes missing words, inappropriate word choice and confusing sentence structures. I am trying to address most of this in my comments.

Response 1: Regarding the comments on the language, the revised manuscript has been edited by MDPI Language Editing Services. We hope the correction will meet with approval.